# Exploring the impact of dexamethasone on gene regulation in myeloma cells

Victor Bessonneau-Gaborit[1,2,*], Jonathan Cruard[1,*], Catherine Guerin-Charbonnel[1,3] , Jennifer Derrien[1],
Jean-Baptiste Alberge[1], Elise Douillard[1,2], Magali Devic[1,2] , Sophie Deshayes[1], Loïc Campion[1,3], Frank Westermann[4,5] ,
Phillipe Moreau[1,2], Carl Herrmann[6], Jérémie Bourdon[7] , Florence Magrangeas[1,2,†] , Stéphane Minvielle[1,2,†]

Among glucocorticoids (GCs), dexamethasone (Dex) is widely used in treatment of multiple myelomas. However, despite a definite benefit, all patients relapse. Moreover, the molecular basis of glucocorticoid efficacy remains elusive. To determine genomic response to Dex in myeloma cells, we generated bulk and single-cell multi-omics data and high-resolution contact maps of active enhancers and target genes. We show that a minority of glucocorticoid receptor-binding sites are associated with enhancer activity gains, increased interaction loops, and transcriptional activity. We identified and characterized a predominant enhancer enriched in cohesin (RAD21) and more accessible upon Dex exposure. Analysis of four gene-specific networks revealed the importance of the CTCF–cohesin couple and the synchronization of regulatory sequence openings for efficient transcription in response to Dex. Notably, these epigenomic changes are associated with cell-to-cell transcriptional heterogeneity, in particular, lineage-specific genes. As consequences, *BCL2L11*-encoding BIM critical for Dex-induced apoptosis and *CXCR4* protective from chemotherapy-induced apoptosis are rather up-regulated in different cells. In summary, our work provides new insights into the molecular mechanisms involved in Dex escape.

## Introduction

Dexamethasone (Dex), a synthetic glucocorticoid (GC) known for its anti-inflammatory and immunosuppressive activities, in combination with immunomodulatory drugs (IMiDs) and proteasome inhibitors (PIs), is the standard induction treatment in transplant-eligible patients with newly diagnosed multiple myeloma (MM). The use of a new class of drugs, the monoclonal antibody daratumumab, in combination with thalidomide (IMiD), bortezomib (PI), and Dex, improved depth of response and progression-free survival (Moreau et al, 2019). Recently, T-cell-engaging therapies administrating bispecific antibodies targeting CD3 and B-cell maturation antigen for patients who had relapsed or refractory MM, in which Dex is delivered in premedication, showed a promising efficacy in depth and duration of the response (Moreau et al, 2022). Despite spectacular therapeutic improvement, few patients are cured; therefore, it is necessary to better understand the precise mechanisms of action of each agent alone or in combination.

Dex exerts its biological functions by binding to the glucocorticoid receptor (GR) encoded by *NR3C1*. Upon Dex binding, the complex translocates to the nucleus, where it associates with DNA at GR-binding sites, acts as a transcription factor (TF), and regulates gene expression (Reddy et al, 2009). GR binding appears to be preprogrammed by the binding of lineage-specific TFs and chromatin accessibility before exposure (Biddie et al, 2011; John et al, 2011). At these loci, GR co-binds with cell-specific pioneer TFs including CEBPB in the liver (Grøntved et al, 2013), PU.1 in the macrophage lineage (Oh et al, 2017), and AP1 in murine hepatocytes (Biddie et al, 2011). GR binds predominantly at distal enhancers (Reddy et al, 2009) and drives transcription by interacting with gene promoters via chromosomal loops. GR binds thousands of locations across the genome but only few enhancers cooperate with each other to activate Dex-responsive genes (Vockley et al, 2016; McDowell et al, 2018). A previous study exploiting protein-directed chromatin interactions approach suggests that at GR-responsive genes, chromatin interaction loops between enhancers and promoters are preestablished, whereas in a subset of genomic loci, GR binding induces de novo interactions (Kuznetsova et al, 2015). High-resolution genome-wide maps of chromatin interactions in response to Dex confirmed and extended the model that GR binding acts predominantly through preestablished chromatin interactions

---

[1]Université de Nantes, CNRS, INSERM, Centre de Recherche en Cancérologie et Immunologie Intégrée Nantes Angers, France   [2]Centre Hospitalier Universitaire, Nantes, France   [3]Institut de Cancérologie de l'Ouest, Nantes, France   [4]Hopp Children's Cancer Center Heidelberg, KITZ, Heidelberg, Germany   [5]Division of Neuroblastoma Genomics, German Cancer Research Center, Heidelberg, Germany   [6]Health Data Science Unit, Medical Faculty Heidelberg and BioQuant, Heidelberg, Germany   [7]Université de Nantes, LS2N, CNRS, Nantes, France

Correspondence: florence.magrangeas@chu-nantes.fr; stephane.minvielle@univ-nantes.fr
*Victor Bessonneau-Gaborit and Jonathan Cruard are co-authors
†Florence Magrangeas and Stéphane Minvielle are co-seniors

and increases their frequency (D'Ippolito et al, 2018). However, these studies did not resolve the influence of increased chromatin accessibility on chromatin loops and enhancer activity as shown by Stavreva et al (2015). Moreover, how these enhancers combine to induce gene expression is still poorly understood. Given that dex is an essential drug in the treatment landscape of MM disease course, the analysis of its molecular action on the genome of myeloma cells and on transcriptional heterogeneity is needed to better understand treatment escape.

# Results

## GR binds to preestablished chromatin sites in MM cells

To investigate the genomic features associated with GR binding in malignant plasma cells, we conducted chromatin immunoprecipitation sequencing (ChIP-seq) for GR, and acetylation of histone H3 at lysine 27 (H3K27ac), assay for transposable-accessible chromatin sequencing (ATAC-seq), and RNA sequencing (RNA-seq) in the Dex-sensitive human myeloma cell line MM.1S exposed to Dex (0.1 $\mu$M) or to equal-volume ethanol (EtOH) for 1 and 4 h (Fig 1A). As expected, a large majority (84%; 15,862/18,844) of GR-binding sites fell within already accessible chromatin and 76% (14,243/18,844) of these sites were annotated as transcriptionally active chromatin at transcription start sites (TSSs) or active enhancers (Fig S1A–C). In addition, GR-binding sites were found in almost all (99%; 808/815) preestablished super-enhancers (SEs) (Hnisz et al, 2013). SE-associated genes included key genes for plasma cell development and MM biology such as *IGLL5*, *IRF4*, *XBP1*, *PRDM1*, and *IKZF1* in line with previous results (Lovén et al, 2013) (Fig S1D).

We next sought to identify TFs that could play an important role in driving GR binding to chromatin in myeloma cells. Motif discovery analysis using the MEME (Multiple Em for Motif Elicitation) (Bailey et al, 2015) algorithm revealed motif enrichment for IRF4 binding (ISRE), a TF critical for MM proliferation and/or survival (Shaffer et al, 2008), and the anticipated GR-responsive element (GRE) in GR-binding sites (Fig 1B). These findings suggest that upon Dex exposure GR binds preferentially to IRF4 chromatin complexes. To test this, Rapid Immunoprecipitation Mass spectrometry of Endogenous protein (RIME) experiments were performed in MM.1S Dex-treated cells using GR as bait (Fig 1C; Table S1). We found that IRF4 was one of the top-ranking partners along with IKZF1 and IKZF3 previously reported to be essential TFs in MM (Krönke et al, 2014). Co-immunoprecipitation (Co-IP) experiments confirmed that GR interacts with IRF4 in the nucleus of MM.1S exposed to Dex (Figs 1D and S1E). In addition, RIME identified known GR interactors including seven subunits of the SWI/SNF chromatin remodeling complex, of which the ATP-dependent remodeling enzyme SMARCA4 (alias BRG1) and ARID1A, required for GR-chromatin remodeling and transcriptional regulation (Fletcher et al, 2002; Trotter & Archer, 2004; Trotter et al, 2008). GR peaks in Dex-treated cells and IRF4 peaks in untreated cells strongly overlapped (73%) with H3K27ac-enriched regions (Fig 1E). Enrichment of the IRF4 motifs and IRF4-binding colocalization with 77% of GR-bound active regulatory regions suggest that IRF4 is the specific TF associated with GR

binding in myeloma cells. However, unlike *CEBPB*, which is strongly Dex-induced in the liver (McDowell et al, 2018), *IFR4* is repressed (log$_2$fold change = −0.55 at 4 h of Dex exposure), which is also the case for its two main responsive genes in MM.1S cells (Low et al, 2019): *TNFRS17* (alias B-cell maturation antigen) and *MANF* (log$_2$fold change = −0.59 and −0.43 at 24 h of Dex exposure, respectively) (Figs 1F and S2).

Despite the large number of GR-binding sites on the genome, only a small number of genes were deregulated in response to Dex (982 up-regulated, 3.2%; 649 down-regulated, 2.1%; abs [log$_2$fold change] > 0.6 and FDR < 0.05) in myeloma cells including well-known ubiquitous GR-responsive genes like *TSC22D3* (alias *GILZ*), *FKBP*5, and cell-specific genes like *BCL2L11* (alias *BIM*), an essential gene for Dex-induced death in MM.1S, and *CXCR*4, the chemokine receptor gene known to be associated with MM progression and poor prognosis (Fig 1F). Thus, suggesting that only a fraction of GR-binding sites is critical for transcriptional activity induced by Dex exposure.

## Enhancer contact map in response to Dex treatment

As expected, GR-bound regions enriched in H3K27ac signal occurred predominantly (85%) at distal enhancers (Fig S3A and Table S2); they were closer to up-regulated genes than to stable genes (Wilcoxon test, $P < 0.0001$) (Fig S3B), closer to each other compared with GR peaks without H3K27ac changes (Wilcoxon test, $P < 0.0001$) (Fig S3C) and formed new SEs associated with known ubiquitous GR-responsive genes, including *DDIT4*, *FKBP5*, and *TSC22D3* (Fig S3D). In an attempt to decipher which GR-binding sites are mandatory to promote transcriptional regulation, we firstly drew an enhancer contact map of GR-activated enhancers and defined at high resolution the changes in chromatin topology resulting from Dex exposure. We employed H3K27ac HiChIP method (Mumbach et al, 2016, 2017) (Fig 2A), which interrogates chromatin contacts between active elements (enhancers or promoters) distal (i.e., located more than 5 kb away from closest TSS) or proximal (i.e., less than 5 kb from closest TSS). We identified 21,249 and 23,278 H3K27ac chromatin interactions across the genome in MM.1S EtOH and MM.1S Dex, respectively. Mapping of the enhancer connectome showed an overall enrichment of H3K27ac ChIP-seq signal in loop anchors compared with non-anchors (Wilcoxon test, $P < 0.0001$) (Fig S4A), a median loop distance of 125 kb (Fig S4B), and revealed that most of the loops (80%) involved a proximal anchor (Fig S4C). We were particularly interested in interaction loops that increased their frequency upon Dex exposure compared with stable and decreased interactions (Fig 2B). The distal interactions subgroup was overrepresented at the expense of the proximal interaction subgroup (Chi2 test, $P < 0.0001$), whereas distal–proximal interactions subgroup remained unchanged (Fig 2C). The anchors were enriched in GR occupancy (Wilcoxon test, $P < 0.0001$) (Fig 2D) and gained chromatin accessibility (Chi2 test, $P < 0.0001$) (Fig 2E). Finally, increased ATAC-seq peaks (27%) in these anchors were enriched in the GRE motif ($P$-val = $1 \times 10^{-30}$), whereas stable ATAC-seq peaks were enriched in the ISRE motif ($P$-val = $1 \times 10^{-69}$). A scanning-motif approach for GRE and ISRE revealed that GRE was found in 11.72% and 1.98% of increased and stable ATAC-seq peaks, respectively. We also found ISRE in 19.48% and 9.68% of increased and stable ATAC-seq peaks, respectively

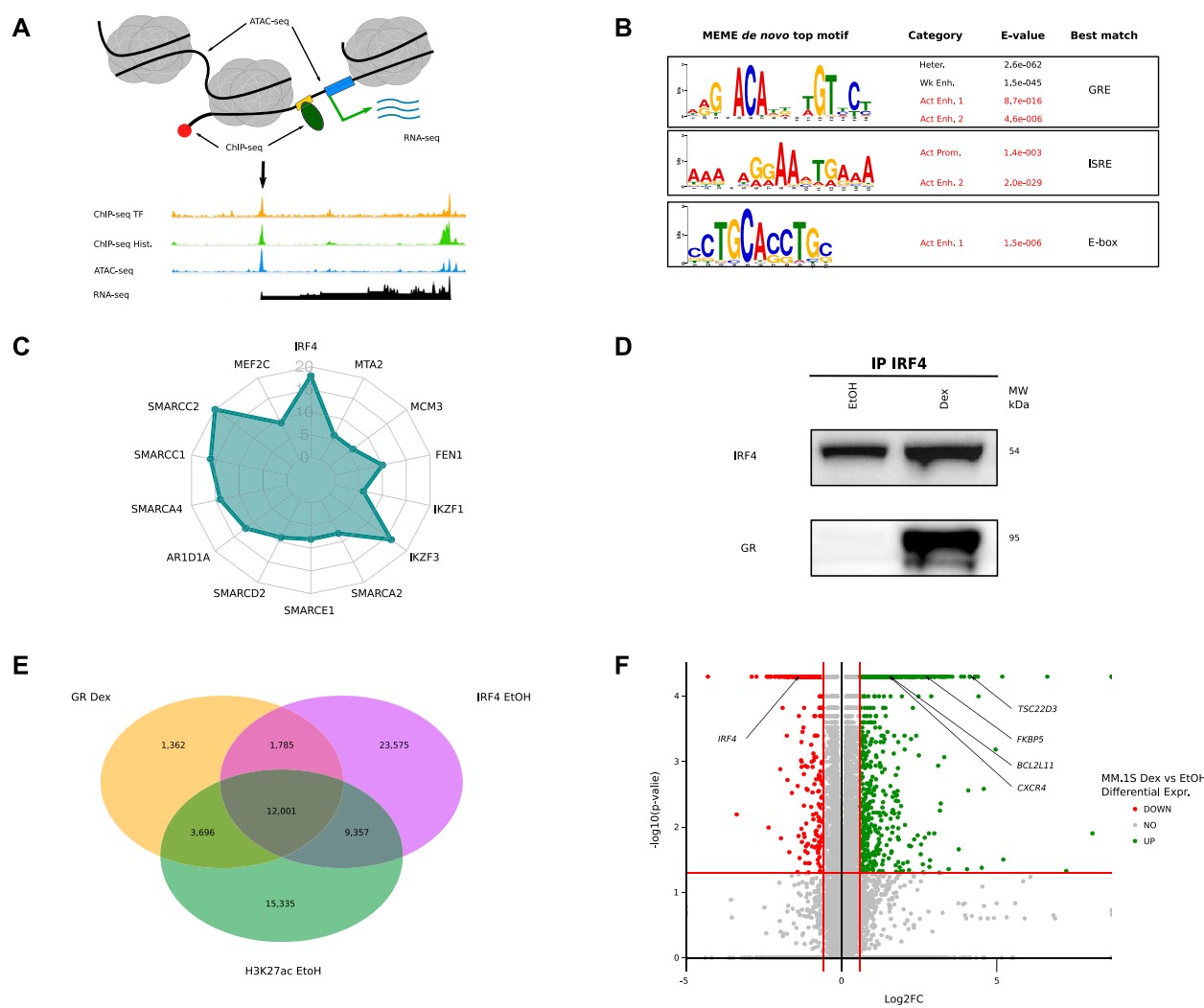

**Figure 1. Preprogrammed chromatin landscape guides GR binding in malignant plasma cells.**
**(A)** Scheme of sequencing data used to define chromatin landscape of MM.1S cell line. **(B)** De novo Multiple Em for Motif Elicitation top motif enriched in MM.1S chromHMM functional states. **(C)** Radar chart showing GR-binding partners using rapid immunoprecipitation mass spectrometry of the endogenous protein method in MM.1S (Dex 1 h); arbitrary units. **(D)** Immunoprecipitation of IRF4 transcription factor in MM.1S cells in control and Dex conditions. Revelation by IRF4 and GR antibodies. **(E)** Overlap of chromatin immunoprecipitation sequencing peaks for GR (MM.1S Dex 1 h), IRF4 (MM.1S EtOH, Lovén et al [2013]), and H3K27ac (MM.1S EtOH). **(F)** Volcano plot of differential RNAseq in Dex treatment compared with control, dysregulated genes are coloured in green (up) and red (down), and key genes are highlighted.

(Fig 2F; Table S3). This reinforces the fact that for an opening to occur, the GR must bind directly to the DNA.

## Predominant enhancer (pE) genomic features

To further investigate the impact of Dex exposure on regulatory networks, we selected among the Dex-increased distal–proximal interaction loops those associated with Dex-up-regulated genes (Fig S5A). We retained 62 interaction loops corresponding to 55 genes (Fig 2B), which displayed high transcriptional variability, in terms of median expression, induction intensity, and ratio of expressing cells (Fig S5B). Half of the genes appeared to be regulated by short-range promoter–enhancer interactions (<5 kb) (Fig S5C). Analysis of the enhancer connectome upon Dex exposure revealed formation of about one SE in 80% of gene networks (Fig

S5D). Of note, the density of interconnections was very varied from one network to another (Fig S5E) but was not associated with the level of gene expression, nor was the number of interactions that increased in the presence of Dex associated with the level of expression (| r | < 0.2). Interestingly, a careful visual inspection of increased interactions across *BCL2L11*, *CXCR4*, *FKBP5*, and *TSC22D3* revealed a particular enhancer that formed more spatial contacts than the others after Dex exposure (Fig S5F). Our observations, together with previous results (Kuznetsova et al, 2015; Vockley et al, 2016), suggest that one particular enhancer could act as a regulatory hub to promote transcription of Dex-responsive genes. To explore this hypothesis, we firstly selected in each network one enhancer (as described in the Materials and Methods section), we retained 34 enhancers referred to as pE (Fig S6A and B; Table S4), among those, 79% were located within a SE. The pEs showed

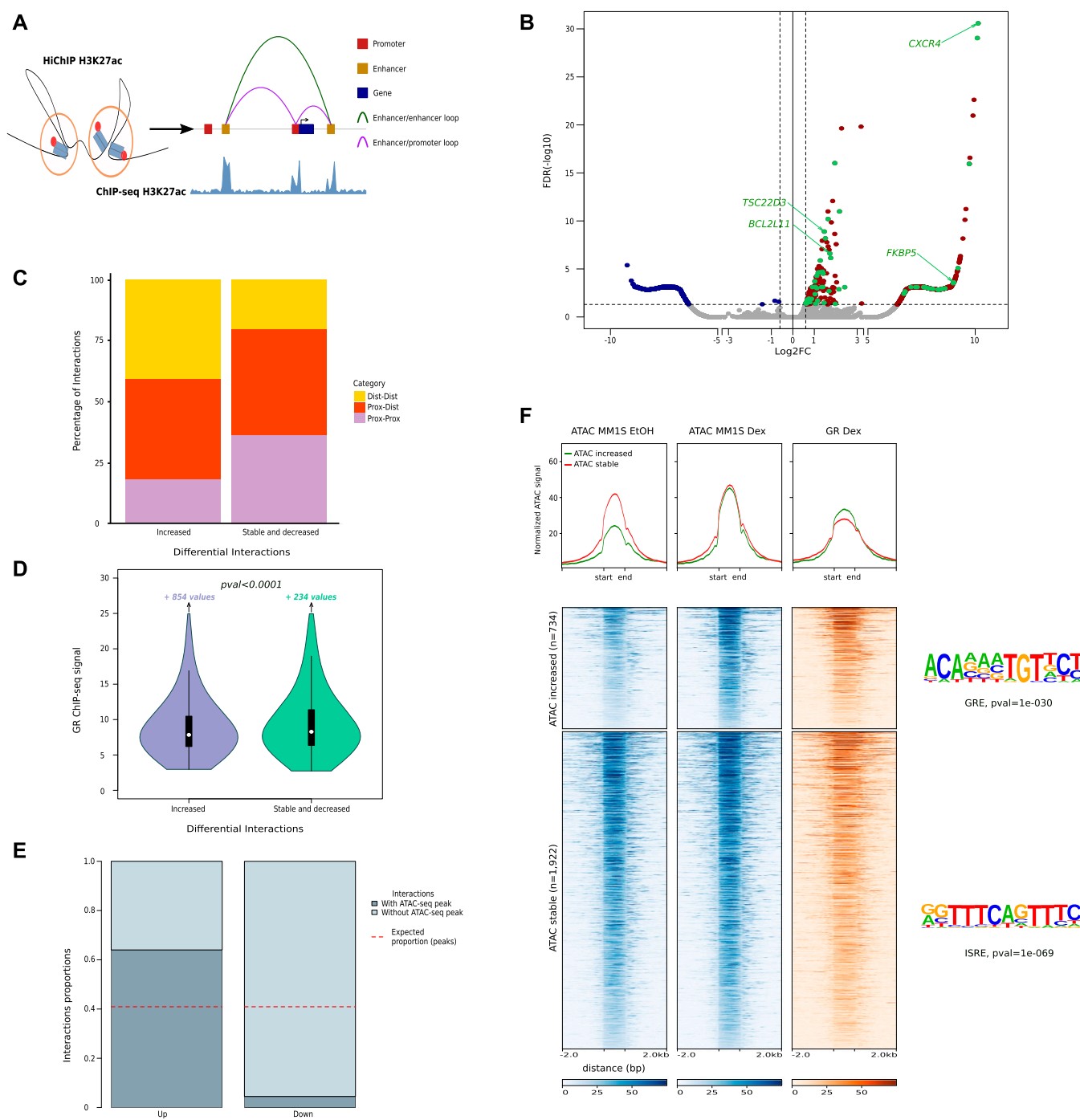

**Figure 2. H3K27 acetylation chromatin interaction in response to Dex.**
**(A)** Scheme depicting H3K27ac loop interactions between regulatory regions through linear genome. **(B)** Volcano plot of HiChIP H3K27ac differentially induced (FDR < 0.05) loops between EtOH and Dex MM.1S cells, increased (log$_2$FC > 0.6) loops in red and decreased (log$_2$FC < −0.6) loops in blue; interactions with anchors overlapping with Dex-induced gene expression are shown with green dots. **(C)** Annotation of interactions depending on three categories: Dist-Dist (>5 kb from closest transcription start site [TSS] for each HiChIP anchors); Prox-Dist (5> kb from closest TSS for one HiChIP anchor and <5 kb for the other); prox-Prox (<5 kb from closest TSS for each HiChIP anchors). **(D)** Box plots illustrating the significantly higher levels of GR signal in increased interactions compared with GR signal in stable or decreased interactions. **(E)** Barplots showing proportion of interactions with and without transposable-accessible chromatin sequencing (ATAC-seq) peaks in increased and decreased interactions. Red dotted lines show the expected proportion of interactions with ATAC-seq peaks in case of no difference between up and down interactions. **(F)** Heatmaps illustrating ATAC-seq signal (EtOH and Dex conditions) and GR chromatin immunoprecipitation sequencing signal in ATAC-seq Dex-increased or Dex-stable peaks located within anchors of the Dex-increased H3K27ac HiCHIP loop. Notable significant motifs found with de novo motif discovery approach are highlighted at right.

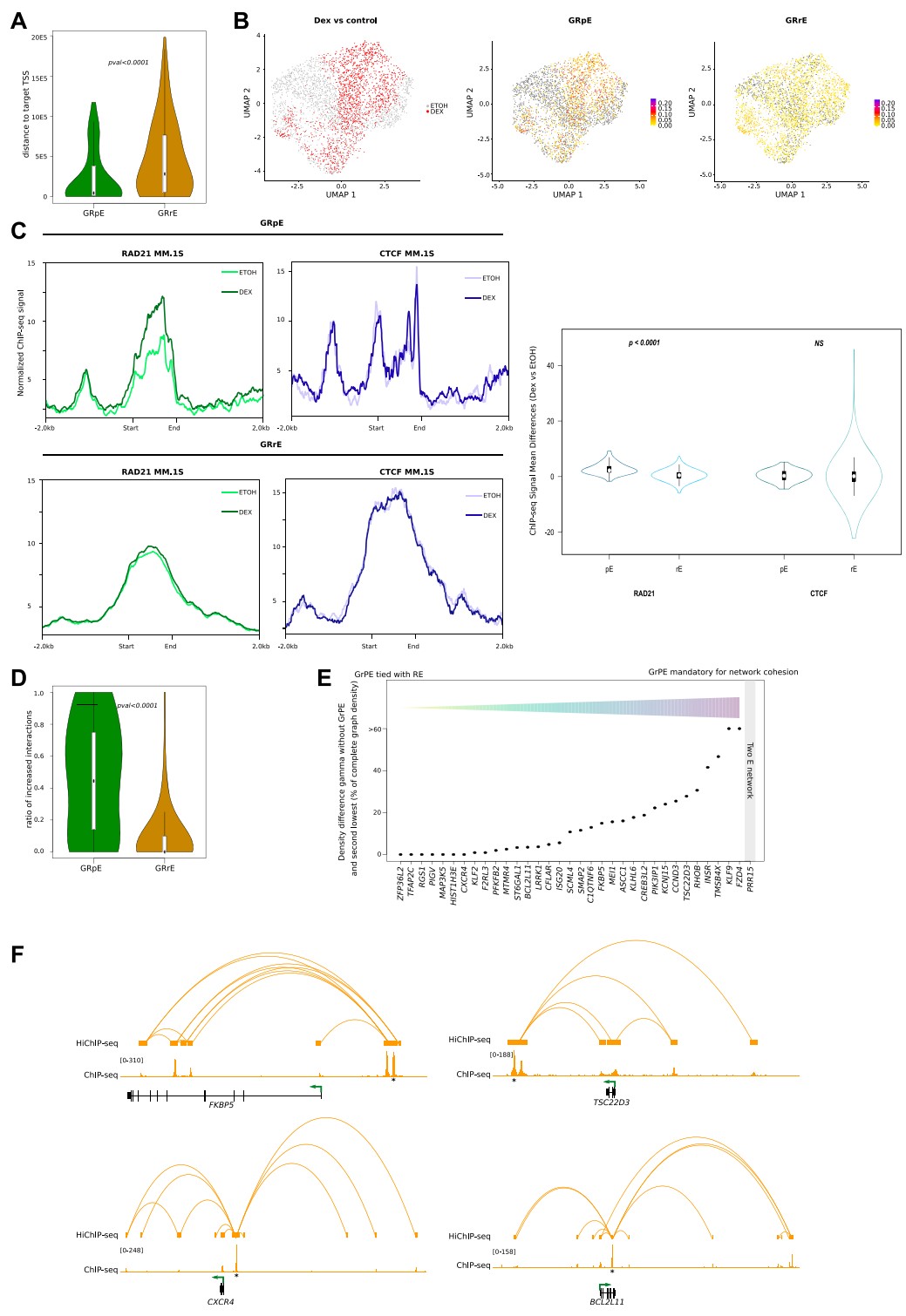

**Figure 3. Comparison of molecular features between pE and rE among regulatory networks.**
**(A)** Box plot illustrating the absolute distance from closest Dex-induced transcription start site of GR predominant enhancers (pEs) (green) and GR regular enhancers (rEs) (yellow). **(B)** Uniform Manifold Approximation and Projection of scATAC-seq profiles in MM.1S cells, coloured by sample of origin (left), GRpE activity score (middle), and GRrE activity score (right). **(C)** chromatin immunoprecipitation sequencing (ChIP-seq) signal for RAD21 (left) and CTCF (right) in control (light green and light blue) and in Dex condition (green and blue) at the GRpE (top) and GRrE positions (bottom); box plots illustrating mean differences between Dex and EtOH ChIP-seq signals for RAD21 (left) and CTCF (right) at GRpE versus at GRrE. **(D)** Box plot illustrating the ratio of Dex-increased chromatin loops for the anchor containing a GRpE (green) or the anchors containing GRrEs (yellow). **(E)** Details of cohesiveness index of pE for each of the 37-gene regulatory networks. The higher the index, the more essential the pE is for the network cohesion. Index could not be computed for PRR15 because the network only has two vertices. **(F)** Snapshot visualisation of GR HiChIP and GR ChIP-seq data at four different loci: *FKBP5*, *CXCR4*, *TSC22D3*, and *BCL2L11*.

different molecular features than regular enhancers (rEs) (Fig 3). The pEs were closer to their target gene promoter (Fig 3A) and more accessible (Figs 3B and S7). We detected an enrichment of cohesin subunit RAD21 after Dex exposure at pEs, which correlated with gene expression (r = 0.31), whereas Dex exposure had no effect on CTCF binding at both pEs and rEs (Fig 3C). In line with our hypothesis, pEs formed more increased interaction loops than rEs (Fig 3D) and were essential for the cohesion of the network (Fig 3E). For example, GR HiChIP experiments, which interrogated chromatin contacts between GR-bound elements, clearly showed the central position of pEs in the *FKBP5*, *TSC22D3*, *CXCR4*, and *BCL2L11* loop networks (Fig 3F). The features of pEs are similar to those of the hub enhancers described by (Huang et al, 2018), but unlike the latter, they regulate other genes than the cell-identity genes. From these results, we conclude that in spite of a great variability of network connections and transcriptional changes, the pEs have common characteristics, they open more, bind more cohesin, and interact more than rEs upon Dex exposure.

## Importance of CTCF–cohesin couple and co-accessibility in Dex response

To characterize the relationship between epigenomic events involving pEs and changes in gene expression, we focused our analysis on the four GR-responsive genes described above: *FKBP5*, *CXCR4*, *TSC22D3*, and *BCL2L11*. These four genes are well representative of the different subgroups of genes regulated by Dex: ubiquitous versus lineage-specific, densely connected versus poorly connected, robust induction versus medium induction. Regarding chromatin interaction maps assessed by H3K27ac-HiChIP experiments, we observed changes among the regulatory network of these genes (Fig 4). At the *FKBP5* gene locus, we found a strong induction of long-distance pE–rE contacts and short-distance rE–rE contacts within SEs in Dex-treated cells compared with EtOH-treated cells (Fig 4A). Increase of contacts between enhancers coincided with strong up-regulation (log$_2$fold change = 4) of *FKBP5*. In contrast, regarding the *CXCR4* gene locus, Dex induced new contacts between pE and *CXCR4* promoter and between the promoters of *CXCR4* and *DARS1* without formation of SE. These new contacts preceded a 2.9-log$_2$fold change increase of *CXCR4* expression, whereas *DARS1* expression remained unchanged (Fig 4B). However, in the case of *TSC22D3* locus, though a strong induction of gene expression after Dex exposure was found (log$_2$fold change = 4.3), interaction loops changes were only moderate (logfold change < 1.5) (Fig 4C). This indicates that the H3K27ac-HiChIP assay is sometimes limited in defining contact maps that could explain gene expression changes, which is probably because of technical constrains. Indeed, in addition to a possible overestimation of interactions, given the high overlap between H3K27ac-increased enhancers and H3K27ac-increased loops (Fig S8A), we cannot exclude, in control conditions, an overrepresentation of labile contacts ineffective for transcription because of the fixation steps that freeze chromatin contacts. In addition, possible artifacts are introduced by bulk analysis, which averages out the effects on heterogeneous populations. To circumvent these limitations, we employed a computational method based on co-accessibility scores at single-cell resolution that predicts *cis*-regulatory DNA interactions between two regulatory elements in the same cell from scATAC-seq data (Pliner et al, 2018) (Fig S8B). As anticipated, Dex

exposure significantly modified co-accessibilities of regulatory elements (P = 0.02; Fig S8C). Analysis of scATAC-seq data at *TSC22D3* gene locus indicated significant gain in accessibility only at GR-bound pE (called GRpE) and the proximal rE (called GR1), both enhancers being clustered in the SE (Fig 5A top). Co-accessibility scores predicted significant Cicero links between these enhancers and the promoter, and within the SE (Fig 5A middle). These results indicate that Dex exposure leads to an increase in the number of cells with co-accessibility between enhancers within SE, and between SE and the promoter. Because we have observed CTCF occupancy at enhancers in control condition and RAD21 enrichment at pEs upon Dex exposure, we investigated binding of these molecules at *TSC22D3* gene locus. We found that the promoter and its SE were in close proximity to convergently oriented CTCF-binding sites predicted to form strong CTCF-associated loops (Oti et al, 2016). Although there were no CTCF occupancy changes, we observed a strong enrichment of cohesin subunit RAD21 at GR-binding sites of GRpE, GR1, and the promoter (Fig 5A bottom). These observations are consistent with a recent report that GR associates with cohesin complex at GR-responsive sites, strengthening preestablished chromatin loops, promoting DNA loop extrusion, and activation of *Tsc22d3* expression (Rinaldi et al, 2022).

We observed a similar phenomenon for the other ubiquitous GR-responsive gene *FKBP5* in response to Dex: an increased accessibility at GR-binding sites of GRpE, GR4, and the promoter (Fig 5B top). However, the number of Cicero-based links within SEs and also between SEs, increased significantly (Fig 5B middle), which was concordant with HiChIP data (Fig 4A). Cicero's links mostly involve GRpE, as shown by its central position in the predicted network (Fig 5B bottom right). We also found RAD21 enrichment at GR-binding sites of two neighbouring enhancers (<4 kb), GRpE and GR6 both located in the downstream SE. In addition, this SE harbored multiple CTCF sites, exhibiting inward-oriented CTCF motifs indicating a CTCF boundary, whereas chromatin configuration at its regulatory partner, the intragenic SE that clustered 4 rEs, exhibited one inward-oriented CTCF-binding motif (Fig 5B bottom). This suggests that GR binding could promote loop extrusion via cohesin mechanism, where a loop anchor (GRpE locus in this study) forms contacts with a contiguous genomic domain, such as SE; those structures are referred to as architectural stripes (Vian et al, 2018) that enhance transcription, in agreement with the recent study of Rinaldi (Rinaldi et al, 2022).

In contrast, the lineage-specific gene *BCL2L11*, already expressed in control condition, displayed a moderate up-regulation upon Dex exposure. We observed that among the 10 regulatory elements bound by GR, only GRpE was more accessible upon Dex exposure (Fig 5C top). Not surprisingly, Cicero's algorithm predicted a new co-accessibility link between the GRpE and the promoter (Fig 5C middle). The network graphs showed the switch of the promoter to a central position in the network directly connected to GRpE upon Dex exposure (Fig 5C bottom right). We also observed an enrichment of cohesin (RAD21) at both GRpE and the promoter which could stabilize contacts, suggesting that GR binding to GRpE promotes GRpE–promoter interactions at the expense of other promoter–enhancer interactions. However, given the moderate increase of transcription, we hypothesize that the "rewiring" occurs in a subset of the cell population. Likewise, in the another lineage-specific gene *CXCR4*, which displayed a mild transcriptional activation upon Dex exposure, we observed a greater accessibility at

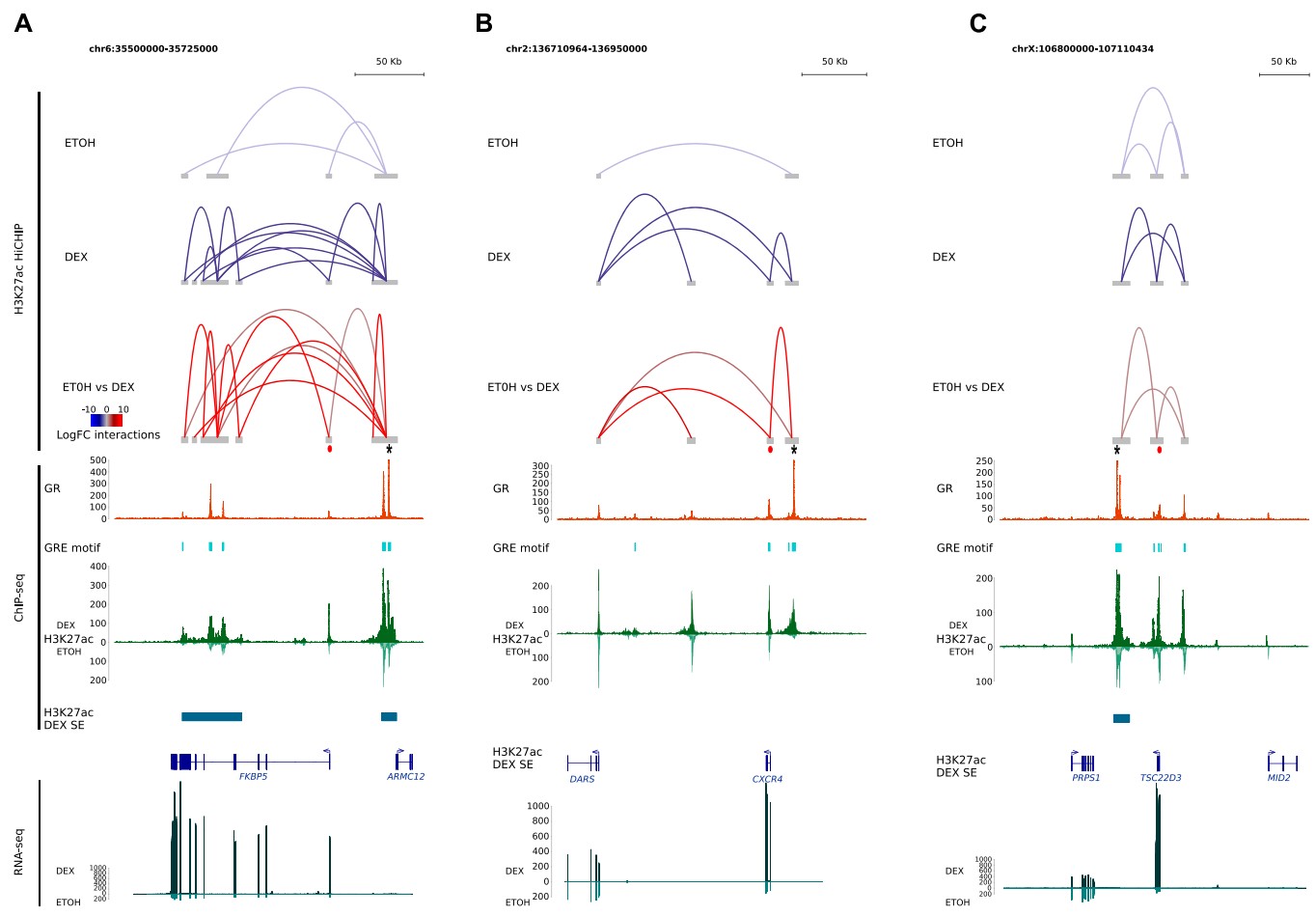

**Figure 4. Epigenetic changes in response to Dex at different loci.**
(A, B, C) Snapshots illustrating the example of GR binding to its consensus motif GR responsive element that increases the H3K27ac chromatin immunoprecipitation sequencing signal as much as the preexistent H3K27ac chromatin interactions for (A) *FKBP5*, (B) *CXCR4*, and (C) *TSC22D3* loci. Predominant enhancer is indicated by a black star, promoter is indicated by a red dot.

GR-binding sites of GRpE and the promoter; though for the latter, it was a small trend (adjusted *P*-value = 0.11) (Fig 5D top). An enlargement of this region clearly revealed a new significant link between GRpE and promoter upon Dex exposure (Fig 5D bottom). Furthermore, the recruitment of RAD21 at GR-binding sites of these looping partners is consistent with a strengthening of GRpE–promoter interaction at least in a subpopulation of cells.

Together, the results show that interaction of the pE with other regulatory sequences (rE and promoter) depends on the organization of the network. Especially, efficient transcription in response to Dex coincides with synchronization of regulatory sequence openings within the cell population and recruitment of cohesin at GR-binding sites, which reinforces the importance of the CTCF–cohesin couple in the stabilization of enhancer–promoter interactions.

### Cell-to-cell transcriptional heterogeneity within myeloma cells after Dex treatment

The importance of synchronized openings suggests that regions that do not open simultaneously may be associated with transcriptional heterogeneity in the cell population. To test this hypothesis, we performed scRNA-seq assay in MM.1S cells collected at 4 and 24 h in the presence of Dex (0.1 $\mu$M) or EtOH. We focused our analysis on the genes most strongly induced by Dex. Analysis of logfold change distribution permitted to isolate 51 highly induced genes, termed single-cell Dex-activated genes (scDAGs) (Fig S9A). More than half (29/51) were in common with the 55 selected genes involved in an increased proximal–distal interaction upon Dex exposure (Fig 2B; Table S4). To measure the level of cell-to-cell transcriptional variability, we calculated scDAG expression information entropy (Landau et al, 2014; Pastore et al, 2019) and found that, among cells exposed to Dex, the median expression increased, leading to an increased fraction of positive cells toward 1; we hence observed a decrease in entropy, that is, less heterogeneity among the cells. However, gene expression among the cells remained heterogeneous, with a high interquartile range (Fig 6A). To further examine the transcriptional variability that remained after Dex exposure, we analyzed the correlations between scDAGs. Although not very strong, we observed higher pairwise correlation coefficients between scDAGs than between random genes (Fig S9B–D). In addition, scDAGs were clustered

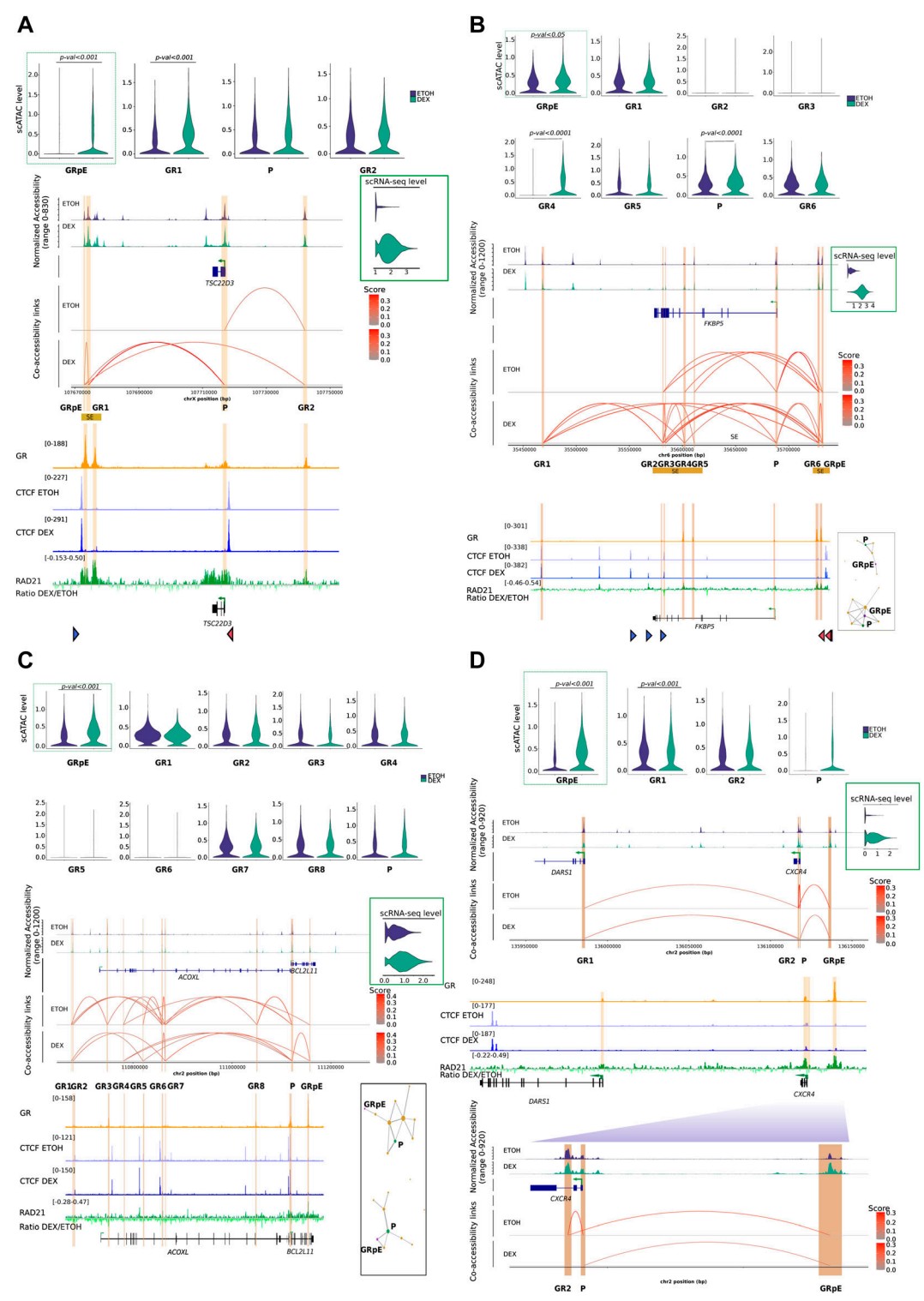

**Figure 5. scMultiome sequencing in response to Dex at 4-gene type Dex-responsive shows different mechanisms.**
**(A)** Violin plot for scATAC-seq data in GR enhancers and promoters in control (EtOH) versus treatment (Dex) (top); snapshot of scATAC-seq data and co-accessibility links at *TSC22D3* (middle), and snapshot of chromatin immunoprecipitation sequencing (ChIP-seq) for GR, CTCF (EtOH and Dex), and RAD21 (Ratio Dex/EtOH) at the corresponding locus (bottom); triangle represents predicted CTCF motif orientation (blue for forward strand and red for reverse strand); Violin plot of scRNA-seq gene expression level for *TSC22D3* (middle right). **(B)** Violin plot for scATAC-seq data in GR enhancers and promoters in control (EtOH) versus treatment (Dex) (top); snapshot of scATAC-seq data and co-accessibility links at *FKBP5* (middle) and snapshot of ChIP-seq for GR, CTCF (EtOH and Dex), and RAD21 (Ratio Dex/EtOH) at corresponding locus (bottom), triangle represents predicted CTCF motif orientation (blue for forward strand and red for reverse strand); Violin plot of scRNA-seq gene expression level for *FKBP5* (middle right); co-accessibility network in EtOH and Dex conditions (bottom right). **(C)** Violin plot for scATAC-seq data in GR enhancers and promoters in control

in two main groups: a large cluster (cluster 1; 45/51 genes), including the proapoptotic gene *BCL2L11* and ubiquitous GR-responsive genes like *TSC22D3*, *FKBP5*, and *DDIT4*, and a second cluster, encompassing six genes, including *CXCR4* (Fig S10A). These results suggest that within Dex-treated cells, two subpopulations coexist: a cell population that predominantly expresses most of scDAGs, referred to as highly Dex-responsive cells, and another population of cells expressing a reduced number of scDAGs. To further explore this possibility, we employed the method recently described by Hoffman et al (2020) to estimate the number of scDAGs expressed in each cell at 4 and 24 h of Dex or EtOH exposure. We found that an EtOH-treated cell had a median background ratio of responding genes (RRG) of 12% (6/51 scDAGs), whereas a Dex-treated cell had a median RRG of 51% (26/51 scDAGs) (Fig 6B), similar to that of 24 h Dex exposure (55%; 21/38 DAGs) (Fig S10B). These results were confirmed by Uniform Manifold Approximation and Projection (UMAP) plots colored by the RRG as a color that revealed an important cell-to-cell heterogeneity among Dex-treated cells. Notably, highly Dex-responsive cells tended to cluster together at the top of the cluster, whereas poorly responsive cells were scattered around (Fig 6C). To know how scDAGs were expressed in the population, we determined the ratio of responding cells (RRC) for each gene. In control condition, the median RRC was 13%, whereas, in the Dex condition, it rose to almost 50% (49%) at 4 h but dropped (37%) at 24 h (Figs 6D and S10C). However, RRC values in the Dex condition were very scattered: if we consider particularly the four genes of interest, the ubiquitous Dex-responsive genes like *FKBP5* and *TSC22D3* exhibited a transcriptional response in almost all cells; in contrast, cell-type specific genes like *BCL2L11* and *CXCR4* were up-regulated in only 38% and 41% of Dex-treated cells, respectively (Figs 6D and S10C). Merged UMAP plots colored according to gene expression of uncorrelated genes *BCL2L11* and *CXCR4* ($R2 = −0.02$) clearly showed that their expressions were mutually exclusive (Fig 6E). The same applied to *CXCR4* and *TSC22D3* and to *CXCR4* and *FKBP5* Fig S11. Altogether, scRNA-seq analysis revealed that, on average, myeloma cells expressed only half of the overall Dex-responsive genes. In addition, in most of the poorly responding cells (i.e., RRG < 50%), *BCL2L11*, the most important GC-induced death gene in MM, was not expressed.

## Discussion

Although the efficacy of Dex in MM can largely be attributed to GR-induced apoptosis, the genomic responses to Dex treatment in malignant plasma cells' genome remain unknown. Given that Dex is used at all stages of treatment, it was crucial to investigate its molecular mode of action by using new genomic tools to better understand treatment escape and provide new insights into combination therapy options. In this study, we confirm the importance of preprogrammed chromatin landscape in guiding most

of GR binding at open and active genomic loci and we show that in plasma cell cellular context, IRF4 is probably the transcriptional factor which cooperates with GR. However, given the suppressive effect of Dex on *IRF4* expression, we cannot rule out the possibility that other factors may be involved after GR binding. As described by Vockley et al (2016), we show that despite a strong association of H3K27ac with GR binding within enhancers engaged in long-range interactions to form lineage-specific networks, the changes in enhancer activity upon Dex exposure are limited and only a small number of genes were deregulated in response to Dex (Fig 1F).

Within some gene-specific regulatory networks, we identified a particular enhancer, referred to as pE, that opens more, interacts more than rEs, and recruits cohesin subunit RAD21 upon GR binding (Fig 3). Targeted genome engineering could be used to directly test the importance of this main enhancer. However, a recent work in acute lymphoblastic leukemia has demonstrated that upon Dex treatment, this specific enhancer promotes an active chromatin interaction with the *BCL2L11* promoter to up-regulate gene transcription (Jing et al, 2018). Similarly, knockdown of a predicted internal enhancer in the *FKBP5* gene locus corresponding to the above GR6 located in the same anchor as the pE induced a 40% reduction in *FKBP5* expression in primary renal proximal tubular epithelial cells (Wilson et al, 2022). We can speculate that activation of both enhancers is essential for a maximal induction of *FKBP5* expression. The activity of critical Dex-regulated enhancers is sensitive to DNA methylation (Jing et al, 2018; Wiench et al, 2011; Wilson et al, 2022), suggesting that inhibition of chromatin accessibility by increased methylation in key regulatory regions as pE may play a critical role in Dex resistance in MM.

We also found that cohesin subunit RAD21 recruitment was associated with Dex-induced GR binding at pEs close to preoccupied CTCF-binding sites, suggesting that CTCF and cohesin are central to mediate stable chromatin loops formed with enhancers activated by GR binding (Fig 5). In addition to stabilizing long-range interactions, association of GR with cohesin complex promotes loop extrusion and long-range gene regulation (Rinaldi et al, 2022). Furthermore, recent results reported the importance of cohesin for regulating the robustness of tissue-specific enhancer–promoter interactions (Aljahani et al, 2022). All these results could explain how GR binding regulates 3D organization of the genome and transcription of both ubiquitous and lineage-specific genes.

It is now established that nucleosome shifts associated with Dex-induced GR binding to both closed nucleosomal or pre-accessible active enhancers require the action of SWI/SNF remodeling complex to regulate gene expression (Johnson et al, 2018). We show that synergistic opening of regulatory sequences within a cell population plays a more important role in transcriptional efficacy than the mere opening of novel regions (Fig 5). Nevertheless, it cannot be excluded that GR at high concentration, induced either by high dose of Dex or prolonged exposure to Dex, may act in a tetrameric state to invade closed chromatin sites and

---

(EtOH) versus treatment (Dex) (top); snapshot of scATAC-seq data and co-accessibility links at *BCL2L11* (middle), and snapshot of ChIP-seq for GR, CTCF (EtOH and Dex) and RAD21 (Ratio Dex/EtOH) at the corresponding locus (bottom); violin plot of scRNA-seq gene expression level for *BCL2L11* (middle right); co-accessibility network in EtOH and Dex conditions (bottom right). **(D)** Violin plot for scATAC-seq data in GR enhancers and promoters in control (EtOH) versus treatment (Dex) (top); snapshot of scATAC-seq data and co-accessibility links at *CXCR4* (middle) and snapshot of ChIP-seq for GR, CTCF (EtOH and Dex), and RAD21 (Ratio Dex/EtOH) at corresponding locus (middle); violin plot of scRNA-seq gene expression level for *CXCR4* (middle right); zoom on *CXCR4* locus (bottom).

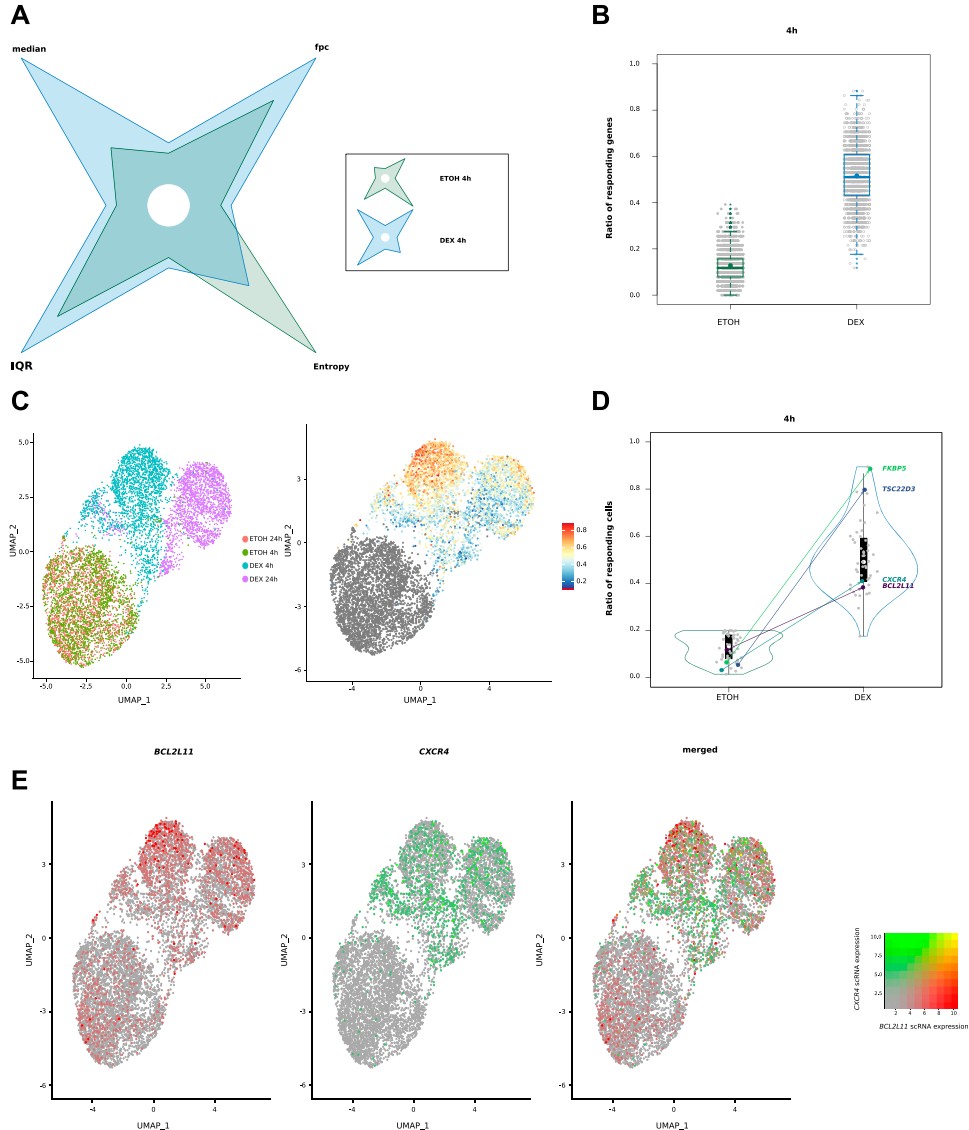

**Figure 6. Cell-to-cell transcriptional heterogeneity after Dex exposure.**
**(A)** Shuriken plot illustrating median, interquartile range, fraction of positive cells, and entropy for control and Dex-treated cells; for each parameter, maximum value is used as reference (1; second value is represented as a proportion of this maximum value). **(B)** Box plots illustrating the ratio of responding genes after 4 h of treatment in control and Dex conditions, with the addition of a dot indicating the mean. **(C)** Uniform Manifold Approximation and Projection (UMAP) plot of MM.1S cells from scRNA-seq coloured by condition (left) and UMAP plot coloured by the ratio of responding genes for 4 h of exposure (right). **(D)** Violon plots illustrating ratio of responding cells after 4 h of treatment. **(E)** UMAP plots coloured for *BCL2L11* expression (red), *CXCR4* (green) expression and merged.

potentially regulate the expression of genes that are not cell-type specific (Paakinaho et al, 2019).

Together, our findings suggest that GR must associate with at least two complexes: SWI/SNF and cohesin to accurately regulate its target genes in myeloma cells.

An important question is whether these multiple enhancer–enhancer and enhancer–promoter interactions at a given gene locus occur in the same cells or not. This is not solved at the moment because of technical limitations. Our work does not answer this question, but provides a computational prediction of the synchronized opening of pairwise anchor sites of interaction loops and gives additional clues in understanding the mechanisms of gene regulation (Fig 5). Our analysis of chromatin architecture, enhancer–promoter interactions, and transcriptional activity in four representative gene loci suggests that Dex-induced GR binding provokes, depending on the gene network, either a rewiring of the promoter–enhancer interactions (Fig S12A), an activation of pE–

promoter interaction (Fig S12B) or a stabilization of the loops inside of a hub (Fig S12C) or not (Fig S12D).

Finally, we showed that these epigenomic changes are associated with a heterogeneous response to Dex in myeloma cells. On average, cells express only 51.6% of scDAGs. Similarly, on average, scDAGs are expressed in only 51.6% of cells after 4 h of treatment, this average decreases slightly to 41.2% at 24 h (Figs 6B and S10B). Expression varies according to genes. In fact, ubiquitous genes like *FKBP5* and *TSC22D3* are expressed in almost all cells, whereas cell-specific genes like *CXCR4* and *BCL2L11* are expressed in less than half of the cells. However, it cannot be ruled out that the differences in expression between the four selected genes may be because of differences in stability or turnover of their mRNAs and cell cycle effect. In myeloma cells, GR binding could lead to a mutually exclusive expression of *BCL2L11* and *CXCR4*, and provides new insights into the mechanisms of drug escape, although considering that GR levels can be a limiting event in Dex treatment (Heuck et al, 2012; Kervoëlen et al, 2015).

Given the potential role of CXCR4 in tumor growth and dissemination (Alsayed et al, 2007; Roccaro et al, 2015), its increased expression upon Dex exposure in a subset of MM.1S cells that do not express the proapoptotic gene *BCL2L11* (Fig 6) raises the provocative possibility that minor populations of myeloma cells could proliferate in response to Dex. In this context, the three-drug combination of a human monoclonal anti-CXCR4 antibody with lenalidomide and Dex or bortezomib and Dex phase Ib/II study demonstrating a high response rate is of particular interest (Ghobrial et al, 2019).

Lastly, we show that IKZF1 and IKZF3 are among the few GR-cobound partners (Table S1), suggesting that these lineage-specific TFs could play a role in Dex response as previously described for the MegaTrans complex in the functionally active estrogen-regulated enhancers (Liu et al, 2014). A recent study showed that these TFs are degraded by IMIds (Sievers et al, 2018). Because both drugs are combined to treat MM patients, we cannot exclude an antagonistic role of these molecules. Further studies to identify the GR-IKZF1/3 target genes, if any, are warranted.

# Materials and Methods

## Molecular biology

### Cell line culture

MM.1S is a multiple myeloma glucocorticoid-sensitive cell line (ATCC CRL-2974). Cells were cultured in RPMI-1640 supplemented with 10% FBS, and 2 mM L-glutamine. Cell line is tested negative for mycoplasma according to the manufacturer's instructions (PCR Mycoplasma-Test Kit I, A9753; ITW Reagent). Cells were initially cultured for 24 h in reduced-serum, hormone-stripped media (RPMI 1640 medium, no glutamine, no phenol red, 32404014; Thermo Fisher Scientific) with 10% charcoal/dextran-treated FBS (Charcoal STRP FBS One Shot, A3382101; Thermo Fisher Scientific) and 2 mM L-glutamine to a concentration of 1 million cells per mL. Subsequently, Dex (D4902; Sigma-Aldrich) was added to the media at 0.1 µM for all treatment timepoints and EtOH was used as vehicle control.

### ChIP-seq procedure

MM.1S cells were exposed to Dex or EtOH for 1 h and crosslinked with freshly made 1% formaldehyde (28908; Thermo Fisher Scientific) for 15 min and quenched with 125 mM Glycine (50046; Sigma-Aldrich) for 10 min. Cells were pelleted and washed in PBS, then pelleted again and stored at −80°C.

ChIP-seq CTCF and RAD21 was performed as previously described (Jin et al, 2018) with the following modifications. Formaldehyde-fixed cells were lysed and chromatin sheared by sonication using a Bioruptor Pico (Diagenode). IP was carried out using 3 µg of polyclonal CTCF antibody (C15410210; Diagenode) or using 5 µg of Anti-Rad21 antibody (ab992; Abcam). DNA from protein-associated complexes and corresponding input samples were washed, eluted, and reversed crosslinking by incubation with RNase A (AM2270; Thermo Fisher Scientific) and protein digested with Proteinase K (25530049; Thermo Fisher Scientific).

Samples were purified with DNA Clean and Concentrator columns (ZD4013; Ozyme) and measured using the Qubit dsDNA HS Kit (Q32851; Thermo Fisher Scientific). Libraries were prepared using NEBNext Ultra II DNA Library Prep according to the manufacturer's instructions (E7103S; New England Biolabs). Libraries were sequenced using Miseq platform (Kit 150cycles V3-PE) with 20 million reads per sample.

ChIP-seq GR (H-300) (sc-8992; Santa Cruz Biotechnology) and H3K27ac (AM-39133; Active Motif) were performed by Active Motif Epigenetic Services. Sequencing depth was 40 million reads for CHIP-seq GR Dex, 38 million reads for CHIP-seq GR EtOH, 28 million reads for CHIP-seq H3K27ac Dex, and 27 million reads for CHIP-seq H3K27ac EtOH.

### RNA-seq procedure

MM.1S cells were exposed to Dex or EtOH for 4 h. Total RNA from MM.1S cells was isolated using direct-zol RNA MicroPrep kits (ZR2060; Ozyme) with on-column DNase treatment according to the manufacturer's instructions. Before RNA-seq, RNA quality was confirmed on the Agilent Bioanalyzer 2100 using the RNA 6000 Nano Kit (5067-1511; Agilent). Total RNA-seq libraries were generated using NEBNext Poly(A) mRNA Magnetic Isolation Module (E7490S; New England Biolabs) and NEBNext Ultra II Directional RNA Library Prep (E7765S; New England Biolabs). Libraries were sequenced using the Illumina HiSeq 2500 (Hiseq Rapid SBS kit v2 2*75 cycles).

### Fast-ATAC procedure

The Fast-ATAC protocol was performed as previously described (Corces et al, 2016) using 0.1 million cells. MM.1S cells were exposed to Dex or EtOH for 1 h, washed in PBS 1X, and centrifuged. The pellet was resuspended in the transposase reaction mix (25 µl of 2x TD buffer, 5 µl of TDE1, 0.5 µl of 1% digitonin, 19.5 µl of nuclease-free water) (FC-121-1030; Illumina, G9441; Promega). Transposition reactions were incubated at 37°C for 30 min in an Eppendorf ThermoMixer with agitation at 1,000 rpm. Transposed DNA was purified using the kit "DNA Clean and Concentrator"-5 (ZD4013; Ozyme). Transposed fragments were amplified and purified as described previously (Buenrostro et al, 2015) with Nextera Index Kit (FC-121-1011; Illumina). qPCR was performed to determine the optimal number of cycles to amplify the library to reduce artifacts associated with saturation PCR of complex libraries. PCR was then performed for the optimum number of cycles using the following PCR conditions: 72°C for 5 min; 98°C for 30 s; and thermocycling at 98°C for 10 s, 63°C for 30 s, and 72°C for 1 min. Libraries were amplified for a total of 11 cycles. Library amplification was followed by solid-phase reversible immobilization methodology (SPRI) size selection to exclude fragments larger than 1,200 bp. Libraries were sequenced using Illumina HiSeq 2500 (Rapid Run HiSeq paired-end 2*75 cycles).

### HiChIP procedure

MM.1S cells were exposed to Dex or EtOH for 1 h, were pelleted, and resuspended in freshly made 1% formaldehyde (28908; Thermo Fisher Scientific) at a volume of 1 ml of formaldehyde for every one million cells. The cells were incubated at room temperature for 10 min with rotation. Glycine (50046; Sigma-Aldrich) was then added

to a final concentration of 125 mM to quench the formaldehyde. The cells were incubated at room temperature for 5 min with rotation. The cells were pelleted and washed in PBS, then pelleted again, and stored at –80°C.

The HiChIP protocol was performed as previously described (Mumbach et al, 2016) using 7.5 µg antibody to H3K27ac (C15410196; Diagenode) with the following modifications. Samples were sheared using Bioruptor Pico (Diagenode), the amount of Tn5 (15027865; Illumina) used and number of PCR cycles performed were based on the post-ChIP Qubit amounts. Libraries were sequenced on NovaSeq 6000 (NovaSeq 6000 S1 Reagent Kit 2*100 cycles).

### Single-cell RNA-seq procedure
For scRNA-seq, MM.1S cells were exposed to Dex or EtOH for 4 and 24 h. Single-cell RNA-seq profiling was performed with the ChromiumTM Single Cell Controller. A total of 6,000 cells was loaded per lane and processed for complementary DNA synthesis and library preparation, per the manufacturer's protocol using 3′ v3.1 chemistry (10X Genomics—1000121). Libraries were sequenced on NovaSeq 6000 (NovaSeq 6000 S1 Reagent Kit 2*100 cycles) to a mean depth of 45,000 reads/cell using the read lengths 26 bp Read1, 8 bp i7 Index, 98 bp Read2.

### Single-cell multiome ATAC + gene expression procedure
For scMultiome, MM.1S cells were exposed to Dex or EtOH for 1 and 4 h. Single-cell 3′ gene expression and open chromatin libraries were simultaneously generated using Chromium Next GEM Single-Cell Multiome ATAC + Gene Expression Kit from 10x Genomics, following the protocol provided by the manufacturer. A total of 5,000 nuclei were loaded per lane on the ChromiumTM Single-Cell Controller. Libraries were sequenced on NovaSeq 6000 (NovaSeq 6000 SP Reagent Kit 2*50 cycles) to a minimum depth of 24,000 reads/nucleus for Gene Expression library and 42,000 reads/nucleus for ATAC library.

### RIME
RIME GR (H-300) (sc-8992; Santa Cruz Biotechnology) and IRF4 (sc-6059; Active Motif) were performed by Active Motif Epigenetic Services. MM.1S cells were exposed to Dex (GR) or EtOH (IRF4) for 1 h and fixed according to the manufacturer's instructions (RIME Cell Fixation protocol, Active Motif). Analyses were performed by an active motif and results are given as a supplementary table (see Table S1). Identified proteins in the IgG negative control were removed and background proteins with a spectral count < 5. The final list is composed of proteins identified in both replicates; their spectral count is the average of spectral count in both replicates.

### Cell protein extraction and fractionation procedure
MM.1S cells were exposed to Dex or EtOH for 1 h. We used NE-PER Nuclear and Cytoplasmic Extraction reagent (78833; Thermo Fisher Scientific) to obtain cytoplasmic and nuclear protein fractions based on the vendor's instructions. Nuclear protein fractions were desalted with ZebaTM Spin Desalting Columns (89882; Thermo Fisher Scientific). Protein was quantified with BC Assay Protein Quantitation Kit (UP40840; Interchim).

### Co-IP
5 µg of anti-IRF4 Antibody (F-4) agarose conjugate (sc-48338; SantaCruz Biotechnology) was added to 50 µg of desalted nuclear protein, the mixture was incubated overnight with mixing at 2–8°C. The immunocomplexes were collected by centrifugation at 1,000g for 5 min at 4°C, washed twice with cold PBS, and resuspended in an electrophoresis sample buffer, 2X (sc-24945; Santa Cruz Biotechnology).

### Western blotting
Proteins extracted from cells or isolated by Co-IP were separated in 4–20% SurePAGE, Bis–Tris Gels (M00655; GeneScript) at 200 V for 30 min, and then transferred onto a nitrocellulose membrane at 150 mA for 2 h. Immunoblotted proteins on the membrane, labelled with specific antibodies were imaged by autoradiography. All primary and secondary antibodies were used according to the manufacturer's instructions. The primary antibodies used were: IRF4 (3E4) (#646402; BioLegend); GR-HRP (sc-393232; Santa Cruz Biotechnology); β-actin (MAB8929; R&D systems); lamin A/C (E-1) (sc-376248; Santa Cruz Biotechnology).

### Computational analysis

### Chromatin state annotation
To obtain functional annotation of MM.1S cell line, we used ChromHMM (v1.11) (Ernst & Kellis, 2010, 2012). Five histone marks available from ENCODE consortium (ENCODE Project Consortium, 2012) (H3K4me1, H3K4me3, H3K27ac, H3K36me3, and H3K27me3) in three different cell lines (MM.1S, U266, and GM12878) were analyzed using hidden Markov model to identify 10 different chromatin states. Default parameters of chromHMM were used. Bam files were binarized into 200-bp genomic windows and the presence or absence of each histone mark was evaluated. Then, we employed biological analysis to annotate those chromatin states giving them biological meanings.

### Treatment of ChIP-seq data
ChIP-seq sequencing quality was assessed with fastqc (v0.11.8) (Andrews, 2010). ChIP-seq read adaptors were firstly trimmed using trimmomatic (v0.39) (Bolger et al, 2014) and then reads were mapped using bowtie2 (v2.1.0) (Langmead et al, 2009) to the Human genome UCSC hg19 (GRCh37) (Kent et al, 2002). Only one mismatch was allowed. After alignment step, unmapped reads, low-quality mapped reads (mapQ < 30), and reads mapped to ENCODE blacklist regions (Amemiya et al, 2019) were removed with samtools (v1.3.1) (Li et al, 2009) for analysis. We also removed reads that were like to be optical and/or PCR duplicates using picard MarkDuplicates (v2.23.5) from GATK (McKenna et al, 2010).

ChIP-seq-enriched regions defined as peaks were called using macs2 (v2.1.1) (Zhang et al, 2008) versus input (sequencing without immunoprecipitation). We only retained peaks higher than specified $P$-value threshold ($P$-val < $1 \times 10^{-7}$).

### Treatment of ATAC-seq data
All ATAC-seq data were processed based on Kundaje laboratory-proposed pipelines (Koh et al, 2016; Liu et al, 2019) available on github. Quality of sequencing assessment, read adaptor

trimming, read mapping to human genome hg19, and filtration were performed the same way as ChIP-seq reads. Before peak-calling steps, and because of the Tn5 insertion, mapped reads were shifted with, respectively, 5 and 4 bp for strand + and strand − with samtools. Finally, enriched regions defined as ATAC-seq peaks were called using macs2 only significant peaks were retained (FDR < 0.05).

### RNA-seq differential analysis

Each RNA-seq sample was mapped using Tophat2 (Trapnell et al, 2009) versus hg19 reference genome. We then employed the proposed protocol (Trapnell et al, 2012) to perform differential expression analysis with cufflinks. Only genes with a LogFC greater than or equal to 0.6 and an FDR < 0.05 were kept for analysis.

### HiChIP data treatment and differential analysis

We employed HiC-pro (Servant et al, 2015) to process HiChIP data from raw data to normalized contact maps. All reads were mapped to hg19 genome using bowtie2 (global parameters: --very-sensitive -L 30 --score-min L, −0.6, −0.2 --end-to-end --reorder; local parameters: --very-sensitive -L 20 --score-min L, −0.6, −0.2 --end-to-end --reorder). Contact maps were generated at different resolutions (1, 2, 15, 20, and 40 kb) and normalized by the iterative correction and eigenvector decomposition method. HiC-pro output directory was then used as input to hichipper (Lareau & Aryee, 2018) with MboI restriction site position for loop calling. Differential analysis of chromatin loops was performed with function exactTest of package edgeR (Robinson et al, 2010), with default parameters except for dispersion, which was set to "trended." Interactions with FDR below 5% and absolute logFC above 0.60 were considered significant.

### Global treatment of genomic data

Genomic data were proceeded using different genomic tools such as Bedtools (v2.28.0) (Quinlan & Hall, 2010) for manipulating genomic files, the homer suite for annotation, and motif scanning (v4.4) (Heinz et al, 2010). Data were also treated using own Python scripts (v2.7).

### Motif search

De novo motif discovery was performed using the MEME suite (v4.11.2) (Bailey et al, 2015) for GR peaks centralized on peak submit and extend with 250 bp in both directions. Motif from 6 to 16 bp were searched with a maximum of five motifs were asked. To identify sequences where a specific motif is found, we employed FIMO tool from MEME. Finally, to identify a centrally enriched motif, we used centriMo from MEME.

### Signal track generation

We employed the bamCoverage tool from the Deeptools (v2.0) (Ramírez et al, 2014) suite to generate bigWig files. Signal track files were normalized using Read Per Genomic Content (RPGC) method also known as the 1X normalization included in bamCovergae options. Once those files were generated, we used the bigwig-Compare tool to create a differential track between H3K27ac with or without Dex. All ChIP-seq and ATAC-seq files were generated with this method. Visualization of signal tracks was obtained using the Integrative Genome Viewer IGV (Robinson et al, 2011).

### Genome ontology analysis

Genome ontology analysis was performed using GREAT (v3.0.0) (McLean et al, 2010) with default parameters (gene regulatory domain: prox. 5 kb upstream and 1 kb downstream; dist. up to 1,000 kb). Enrichment statistics were computed using binomial and hypergeometric gene-based tests. Pathways were selected as significantly enriched if the false discovery rate (FDR q-value) was lower than 0.01.

### Differential analysis of ChIP-seq H3K27ac peaks

To find H3K27ac ChIP-seq responding to GR binding, we first selected all H3K27ac peaks found within GR peaks (n = 16,228). On those sites, we then estimated the normalized count (RPGC) of H3K27ac ChIP-seq in both conditions. $Log_2$fold changes were then calculated for each site and we consider H3K27ac Dex increased all sites with a $log_2$FC higher than 0.1. The H3K27ac Dex-increased peaks are given in the supplementary table (see Table S2).

### Identification of the GRpE among each regulatory network of Dex-responsive genes

It has been shown that GR-binding sites with effects on expression are activated and enriched for the GR-binding motif (Vockley et al, 2016; McDowell et al, 2018). Inspecting more particularly specific networks: for example, *TSC22D3* and *BCL2L11* (Fig S6A and B), it appeared that activated enhancers with the GR-binding motif also displayed a strong GR signal, with the value within the 1% highest values on the whole genome. Based on these findings, we defined pE as follows: pE had to overlap with the presence of a GR motif and have a not-negative Dex/EtOH $log_2$ ratio for H3K27ac signal; among the peaks satisfying those conditions, the one with the highest GR signal value was selected, provided that this value was at least 80% of the maximum GR signal for the region (see Table S4).

### Differential analysis of ATAC-seq peaks found within chromatin loop anchors

We collected all ATAC-seq peaks found within chromatin loop anchors and, for each peak, we estimated the RPGC count of ATAC-seq in EtOH and Dex conditions. $Log_2$FC was then estimated and all ATAC-seq peaks with a LogFC greater than or equal to 0.6 were considered as ATAC up. The ATAC Dex-increased peaks obtained are given as supplementary table (see Table S3).

### CTCF motif orientation-based loop prediction

To predict which CTCF ChIP-seq peaks could potentially create DNA contact, we employed the algorithm and scripts proposed by Oti et al (2016) on our own CTCF ChIP-seq data in MM.1S cell line to define predicted CTCF loops based on motif orientation.

### Global treatment of single-cell data

Preprocessing steps for single-cell data were done using CellRanger Software suite, respectively, cellranger (v5.0.0) (Zheng et al, 2017) and cellranger-arc (v1.0.1) for scRNA-seq and scMultiome-seq (Satpathy et al, 2019). For both types of data, the hg38 genome assembly provided by 10xGenomics was used

for alignment. Further analyses were performed on R (v3.6). For scRNAseq, Count matrices were loaded into R using the Seurat package (v3.9.9) (Satija et al, 2015). For each cell, we calculated the percentage of mitochondrial reads (percent.mt) and the percentage of nuclear-retained lncRNA (percent.nc). We also used the CellCycleScoring function from Seurat to assign a cell cycle state to each cell (Phase); the assignment of the cell cycle state is based on the S. score and G2M.score calculated by this function. Cells were then filtered on the following criteria: 5 < percent.mt < 25, percent. nc < 10, a minimum of 2,000 reads, and 1,500 different genes expressed. Normalization was performed using the Seurat NormalizeData function with standard parameters. The function FindVariableFeatures was then used to select the 3,000 most variable features, those features have been scaled with Seurat ScaleData function, because cell cycle was a major part of the variability, we added S.score and G2M.score to the vars. to.regress argument of the function. We reduced dimension using RunPCA from Seurat; only the first 30 dimensions were used for downstream analyses. We also calculated a 2D embedding of our cells with RunUMAP; neighbor search and clustering were performed using FindNeighbors and FindClusters functions, with default parameters. For scMultiome-seq, RNA and ATAC matrices were loaded into R using Seurat and Signac (v1.1.0) (Stuart et al, 2020 Preprint) packages. For each cell, we calculated the percent. mt, percent. nc and the Phase. For the ATAC data, we also calculated TSS score and the ratio of reads overlapping with the blacklisted regions of the genome contained in the blacklist hg38 unified provided by the Signac package. Cells detected by cellranger were filtered on both RNA and ATAC data. For RNA data, we kept cells between 3,800 and 150,000 reads and more than 2,000 different genes expressed. We also kept cells with a percent. mt between 5 and 30 and a percent. nc lower than 8. For ATAC data, we kept cells with a number of reads between 10,000 and 500,000 and a number of different features between 5,000 and 60,000. We also filtered cells with a TSS enrichment between 3.5 and 15, a nucleosome signal lower than 1.5 and more than 50% reads in peaks. For normalization and dimensionality reduction, we used the RunTFIDF and RunSVD functions from the Signac packages. RunSVD was run on the features selected by FindTopFeatures with min.cutoff set to q80. UMAP embedding, neighbor search, and clustering were performed the same way as for the RNA data alone. Differential expression and accessibility were tested using the findMarkers function provided by Seurat, with test. use argument, respectively, set to "MAST" and "LR."

### Assessment of scATA-seq peaks co-accessibility

Co-accessibility scores between scATACseq peaks were calculated using Cicero (v1.3.4.11) (Pliner et al, 2018) with default parameters. Co-accessibility tables were built on each treatment condition separately. From those tables of co-accessibility scores, we built two networks for each of the 34 Dex-responsive genes GRpEs on one hand and GRrEs on other hand. Nodes of those networks were defined as all peaks overlapping with the selected regions. Edges were built using the co-accessibility tables, considering only connections with a score higher than 0.1.

### scDAGs

We studied the distribution of logFC above 0 and found out it was bimodal with a small part of positive logFC being far from the main part. We then used Gaussian mixture model to identify the small subpopulation of high logFC genes, that is, scDAGS.

### RRG

For each gene, we computed the 80th percentile of expressed values in untreated cells; we then calculated, for each cell, the RRG as the percentage of scDAGs with expression value above the gene threshold for untreated cells, 4 h of Dex-treated cells, and 24 h Dex-treated cells.

## Data Availability

ChIP-seq, HiChIP-seq, ATAC-seq, RNA-seq, scRNAseq, and scMultiome have been deposited at the European Genome-phenome Archive (EGA, https://www.ebi.ac.uk/ega), which is hosted by the EBI and the CRG, under dataset accession EGAD00001011136, EGAD00001011138, EGAD00001011135, EGAD00001011137, EGAD00001011139, and EGAD00001011140, respectively.

## Supplementary Information

## Acknowledgements

We thank the Genomics and Bioinformatics core facility of Nantes (Geno-BiRD, Biogenouest, IFB) for its technical support. We thank the Fondation Française Pour la Recherche contre le Myélome et les gammapathies monoclonales (FFRMG), the Programme d'investissements d'Avenir I-SITE NexT (ANR-16-IDEX-0007) the Pays de la Loire, the SIRIC ILIAD (INCa-DGOS-Inserm-12558), and Celgene for supporting this study.

### Author Contributions

V Bessonneau-Gaborit: data curation, formal analysis, and writing—original draft, review, and editing.
J Cruard: data curation, formal analysis, and writing—original draft, review, and editing.
C Guerin-Charbonnel: data curation, formal analysis, and writing—original draft, review, and editing.
J Derrien: data curation, formal analysis, and writing—original draft, review, and editing.
J-B Alberge: data curation.
E Douillard: data curation.
M Devic: data curation.
S Deshayes: data curation.
L Campion: formal analysis.
F Westermann: data curation.
P Moreau: data curation and funding acquisition.

C Herrmann: data curation, formal analysis, and writing—original draft, review, and editing.

J Bourdon: data curation, formal analysis, and writing—original draft, review, and editing.

F Magrangeas: conceptualization, data curation, formal analysis, supervision, investigation, methodology, and writing—original draft, review, and editing.

S Minvielle: conceptualization, supervision, validation, investigation, methodology, project administration, and writing—original draft, review, and editing.

## Conflict of Interest Statement

The authors declare that they have no conflict of interest.

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
