## [Reviewer comments · Life Science Alliance]

Exploring the Impact of Dexamethasone on Gene Regulation in Myeloma Cells

Victor Bessonneau-Gaborit, Jonathan CRUARD, Catherine GUERIN-CHARBONNEL, Jennifer DERRIEN, Jean-Baptiste ALBERGE, Elise DOUILLARD, Magali DEVIC, Sophie DESHAYES, Loic Campion, Frank Westermann, Phillipe MOREAU, Carl HERRMANN, Jeremie BOURDON, Florence MAGRANGEAS, and Stephane Minvielle

DOI: <https://doi.org/10.26508/lisa.202302195>

Corresponding author(s): Stephane Minvielle, University of Nantes, CRCI2NA, Nantes; CHU Nantes, France and Florence MAGRANGEAS, University of Nantes, CRCI2NA, Nantes; CHU Nantes, France

Review Timeline:

Submission Date:	2023-06-01
Editorial Decision:	2023-06-02
Revision Received:	2023-06-20
Editorial Decision:	2023-06-20
Revision Received:	2023-06-23
Accepted:	2023-06-26

Transaction Report:

Please note that the manuscript was previously reviewed at another journal and the reports were taken into account in the decision-making process at *Life Science Alliance*. Since the original reviews are not subject to Life Science Alliance's transparent review process policy, the reports and author response cannot be published.

June 2, 2023

Re: Life Science Alliance manuscript #LSA-2023-02195-T

Victor Gaborit
University of Nantes, CRCI2NA; CHU Nantes, France

Dear Dr. Gaborit,

Thank you for submitting your manuscript entitled "Exploring the Impact of Dexamethasone on Gene Regulation in Myeloma Cells: An Analysis of Alterations in Enhancer-Promoter Connectivity and Transcriptional Heterogeneity" to Life Science Alliance. We invite you to submit a revised manuscript addressing the Reviewer points

- Address Reviewer 1's comments
- Address Reviewer 2's Major comment #2, and the Minor comments.
- Address Reviewer 3's remaining comments.

Thank you for this interesting contribution to Life Science Alliance. We are looking forward to receiving your revised manuscript.

Sincerely,

Eric Sawey, PhD
Executive Editor
Life Science Alliance
<http://www.lsa-journal.org>

- A letter addressing the reviewers' comments point by point.
- An editable version of the final text (.DOC or .DOCX) is needed for copyediting (no PDFs).
- High-resolution figure, supplementary figure and video files uploaded as individual files: See our detailed guidelines for preparing your production-ready images, <https://www.life-science-alliance.org/authors>
- Summary blurb (enter in submission system): A short text summarizing in a single sentence the study (max. 200 characters including spaces). This text is used in conjunction with the titles of papers, hence should be informative and complementary to the title and running title. It should describe the context and significance of the findings for a general readership; it should be written in the present tense and refer to the work in the third person. Author names should not be mentioned.
- By submitting a revision, you attest that you are aware of our payment policies found here: <https://www.life-science-alliance.org/copyright-license-fee>

B. MANUSCRIPT ORGANIZATION AND FORMATTING:

June 20, 2023

RE: Life Science Alliance Manuscript #LSA-2023-02195-TR

Dr. Stephane Minvielle
University of Nantes, CRCI2NA, Nantes; CHU Nantes, France
8 quai moncoussu
Nantes 44000
France

Dear Dr. Minvielle,

Thank you for submitting your revised manuscript entitled "Exploring the Impact of Dexamethasone on Gene Regulation in Myeloma Cells". We would be happy to publish your paper in Life Science Alliance pending final revisions necessary to meet our formatting guidelines.

- please add ORCID ID for the corresponding and secondary corresponding author--you should have received instructions on how to do so
- please add the Twitter handle of your host institute/organization as well as your own or/and one of the authors in our system
- please add your main, supplementary figure, and table legends to the main manuscript text after the references section
- please correct the call out for Figure 6F as it doesn't have Panel F, but there is a missing call out for Figure 6E
- please add callouts for Figure S125A-D to your main manuscript text
- you may consider uploading Figure S12 as a Graphical Abstract instead, but this is up to you

A. FINAL FILES:

B. MANUSCRIPT ORGANIZATION AND FORMATTING:

Sincerely,

June 26, 2023

RE: Life Science Alliance Manuscript #LSA-2023-02195-TRR

Dr. Stephane Minvielle
University of Nantes, CRCI2NA, Nantes; CHU Nantes, France
8 quai moncoustu
Nantes 44000
France

Dear Dr. Minvielle,

Thank you for submitting your Research Article entitled "Exploring the Impact of Dexamethasone on Gene Regulation in Myeloma Cells". It is a pleasure to let you know that your manuscript is now accepted for publication in Life Science Alliance. Congratulations on this interesting work.

DISTRIBUTION OF MATERIALS:

Again, congratulations on a very nice paper. I hope you found the review process to be constructive and are pleased with how the manuscript was handled editorially. We look forward to future exciting submissions from your lab.

Sincerely,
